# A next-generation tumor-targeting IL-2 preferentially promotes tumor-infiltrating CD8+ T-cell response and effective tumor control

Zhichen Sun [1,2], Zhenhua Ren [3], Kaiting Yang[1,2], Zhida Liu [3], Shuaishuai Cao[1,2], Sisi Deng[2], Lily Xu[4], Yong Liang[1], Jingya Guo[1,2], Yingjie Bian [1], Hairong Xu[1], Jiyun Shi[5], Fan Wang[5,6], Yang-Xin Fu [3] & Hua Peng [1]

While IL-2 can potently activate both NK and T cells, its short in vivo half-life, severe toxicity, and propensity to amplify Treg cells are major barriers that prevent IL-2 from being widely used for cancer therapy. In this study, we construct a recombinant IL-2 immunocytokine comprising a tumor-targeting antibody (Ab) and a super mutant IL-2 (sumIL-2) with decreased CD25 binding and increased CD122 binding. The Ab-sumIL2 significantly enhances antitumor activity through tumor targeting and specific binding to cytotoxic T lymphocytes (CTLs). We also observe that pre-existing CTLs within the tumor are sufficient and essential for sumIL-2 therapy. This next-generation IL-2 can also overcome targeted therapy-associated resistance. In addition, preoperative sumIL-2 treatment extends survival much longer than standard adjuvant therapy. Finally, Ab-sumIL2 overcomes resistance to immune checkpoint blockade through concurrent immunotherapies. Therefore, this next-generation IL-2 reduces toxicity while increasing TILs that potentiate combined cancer therapies.

[1] Key Laboratory of Infection and Immunity of CAS, Institute of Biophysics, Chinese Academy of Sciences, Beijing 100101, China. [2] University of Chinese Academy of Sciences, Beijing 100049, China. [3] Department of Pathology, University of Texas Southwestern Medical Center, Dallas, TX 75235−9072, USA. [4] Department of Biological Sciences, University of Wellesley College, Wellesley 02481 MA, USA. [5] Key Laboratory of Protein and Peptide Pharmaceuticals, CAS Center for Excellence in Biomacromolecules, Institute of Biophysics, Chinese Academy of Sciences, Beijing 100101, China. [6] Medical Isotopes Research Center and Department of Radiation Medicine, School of Basic Medical Sciences, Peking University, Beijing, China. Correspondence and requests for materials should be addressed to Y.-X.F. (email: yang-xin.fu@utsouthwestern.edu) or to H.P. (email: hpeng@moon.ibp.ac.cn)

nterleukin-2 (IL-2), a T-cell growth factor induced by antigen stimulation, is a pleiotropic cytokine that plays pivotal roles in the immune response[1,2]. As a potent inducer of cytotoxic T cells and NK cells, IL-2 was one of the first FDA-approved immunotherapy drugs for metastatic melanoma and renal cell cancer[3,4]. Unfortunately, IL-2 immunotherapy has not been widely applied due to its short half-life in vivo and severe toxicity at the therapeutic dosage[5–7]. In addition, IL2 induces proliferation of regulatory T cells (Tregs) by binding to IL-2 receptor alpha (IL-2Ra), which is preferentially expressed on Tregs[8–10]. Depletion of Treg cells has been shown to enhance IL-2-induced antitumor immunity, suggesting that Treg may be a major barrier for IL-2-mediated CTL expansion[11].

IL-2 exerts stimulatory and regulatory functions by binding to various IL-2Rs, including monomeric, dimeric, and trimeric IL-2Rs. Some T cells, such as Tregs, express high-affinity heterotrimeric receptors composed of CD25 (IL2Ra), CD122 (IL2Rb), and CD132 (the common cytokine receptor γ chain) subunits. In contrast, naïve CD8 T cells, CD4/CD8 memory T cells, and NK cells express a lower-affinity dimeric receptor, which lacks the CD25 subunit[2,12]. Some strategies introduce specific mutations to IL-2 (IL-2 muteins) to induce preferential binding to either CD25 or CD122, thus stimulating specific immune cell subsets. An IL-2 mutein with decreased affinity to CD25 (mainly expressed on pulmonary endothelial cells and Treg cells) was developed to possess more effective tumor therapeutic effects than the wild-type IL-2[13,14]. The superkine, an IL-2 mutein with increased CD122 affinity, can preferentially expand CTL but also remain to increase Treg[15].

In order to limit systemic toxicity, antibody-based delivery of wild-type (WT) IL-2 (Ab-IL2) has been studied[16–21]. However, the higher affinity of IL-2 over the antibody might actually limit tumor targeting of Ab-IL2. Several recent studies show that immunocytokine efficacy and biodistribution of IL-2 variants may not be sufficiently attributed to tumor-antigen targeting[22,23]. Systemic delivery of Ab-IL2 may likely activate T cells in lymphoid and nonlymphoid tissues that contribute to severe toxicity and limited antitumor efficacy.

The increase of tumor-infiltrating lymphocytes (TILs) is an important biomarker for predicting responses to PD-L1 blockade therapy[24]. The functions of IL-2 therapy are known to directly activate CTLs in lymphoid and nonlymphoid tissues[17,25], but it is not yet clear whether the preexisting T cells inside the tumor are sufficient for the effectiveness of IL-2 therapy. While most recent studies have focused on checkpoint blockade[14,26], more attention should be paid to the increased Tregs inside the tumor microenvironment (TME) that limit IL-2 binding to and subsequent stimulation of CTLs.

Targeted therapy with epidermal growth factor receptor-tyrosine kinase inhibitor (EGFR-TKI) has been approved as the first-line treatment of EGFR mutation-positive cancer. Despite high objective response rates, most patients exhibit tumor relapse several months after treatment[27]. Similarly, anti-Her2/neu has been approved by the FDA for treating patients with high Her2/neu expression in tumor tissues[28]. Whether IL-2 therapy can limit such relapse has not been tested. In this study, we generate tumor-targeting Ab-sumIL-2 with decreased CD25 binding while increased CD122 binding that allows more efficient CTL-targeting inside the TME. We have further discovered that Ab-sumIL2, a potent recombinant immune immunocytokine, could synergize with TKI treatment, operative therapy, and checkpoint blockades for more effective CTL expansion to improve the complete response rate and to limit tumor relapse.

## Results

### High Treg infiltration might limit the therapeutic effects of IL-2. We have shown that the number of Treg cells increase as the

tumors progress in various tumor models[29]. Tregs express high-affinity receptors that can absorb and utilize IL-2 much faster than do effector T cells[30]. We wondered whether low level production of IL-2 might further limit IL-2 from interacting with and stimulating CTL. We first assessed the association between IL-2 expression in the tumor and the clinical outcome using TIMER website analysis[31]. High expression of IL-2 was associated with good clinical outcome in melanoma (Supplementary Fig. 1a). Altogether, these data demonstrate that a lack of IL-2 might contribute to limited CTL responses in the tumor.

On the basis of these results, we investigated if an additional supply of IL-2 might be helpful for tumor control. We first treated a B16F10 tumor with an FDA-approved free IL-2 (Fig. 1a). The B16F10 tumor growth could not be controlled by a low-dose-injection of commercial free IL-2 that was applied every other day for a total of three injections. Considering the very short half-life of free IL-2, we generate an IL-2 fusion protein fused to fragment crystallizable (Fc) region of human immunoglobulin G1 to stabilize IL-2 and increase its half-life. While the same molar of IL-2-Fc resulted in better tumor control than free IL-2, tumors become resistant to the treatment in 1–2 weeks. This indicates the presence of negative factors within the tumor, limiting the therapeutic effects of IL-2. Indeed, we found a much higher percentage of Treg cells in TILs than in the lymphoid organs (Fig. 1b, Supplementary Fig. 1b, c). We also analyzed the expression level of CD25 on different subtypes of T cells in the cancer patients based on the single-cell RNA-sequencing data from Zhang's group[32]. The result shows that the expression level of CD25 on Treg cells was much higher than effector T cells in cancer patients (Supplementary Fig. 1d). It consequently suggests that the presence of Treg cells inside the tumor limits the therapeutic effects of IL-2.

### SumIL-2 preferentially binds to CD8⁺ T cells but not to Treg cells, resulting in improved tumor control. It has been reported that F42A in IL-2 can significantly reduce its binding to IL-2R alpha[14]; while super IL-2 contains L80F, R81D, L85V, I86V, and I92F mutants that can increase its binding to IL2R beta[15]. IL2-Fc has previously been used to enhance IL-2 activity by extending its half-life[23]. We first tested the function of F42A mutant IL-2; the data showed that F42A mutant IL-2 could not effectively control tumor growth (Supplementary Fig. 2a). To improve the function of IL-2 in stimulating CTL, we linked Fc portion to IL-2 and introduced F42A, L80F, R81D, L85V, I86V, and I92F mutations to make the super mutant IL2-Fc (abbreviated as sumIL-2-Fc), which is a stable IL-2 functioning as a preferential activator for effector T cells but not Treg cells (Supplementary Fig. 3a). We first compared the affinity of WT and different mutant IL-2 to the activated CD8 T cells (CD3$^+$CD8$^+$CD44$^{hi}$) and Treg cells (CD3$^+$CD4$^+$Fop3$^+$) via using flow cytometry. When mouse spleen cells were incubated with IL-2 variants, WT IL-2 has a higher affinity to Treg cells than to activated CD8$^+$ T cells; F42A mutant IL-2 which has decreased affinity to CD25 does reduce its binding to Treg cells, while it does not bind to the activated CD8$^+$ T cells effectively either. Super IL-2 has a high binding ability to both cell types. However, only our sumIL-2 favorably binds to the activated CD8$^+$ T cells but not Treg cells (Fig. 1c, Supplementary Fig. 3b). To further compare the effect of these IL-2 variants on the expansion of activated CD8$^+$ T cells and Treg cells in vivo, WT C57BL/6 mice were treated with WT IL-2 and IL-2 variants respectively, activated CD8$^+$ T cells and CD4$^+$ Treg cells from the spleen were analyzed by flow cytometry. We found that sumIL2-Fc induced a higher number of activated CD8 T cells than WT IL2-Fc, meanwhile, presents weaker ability to induce Treg cells (Fig. 1d). Thus, a higher activated CD8 T cells/Treg ratio was also

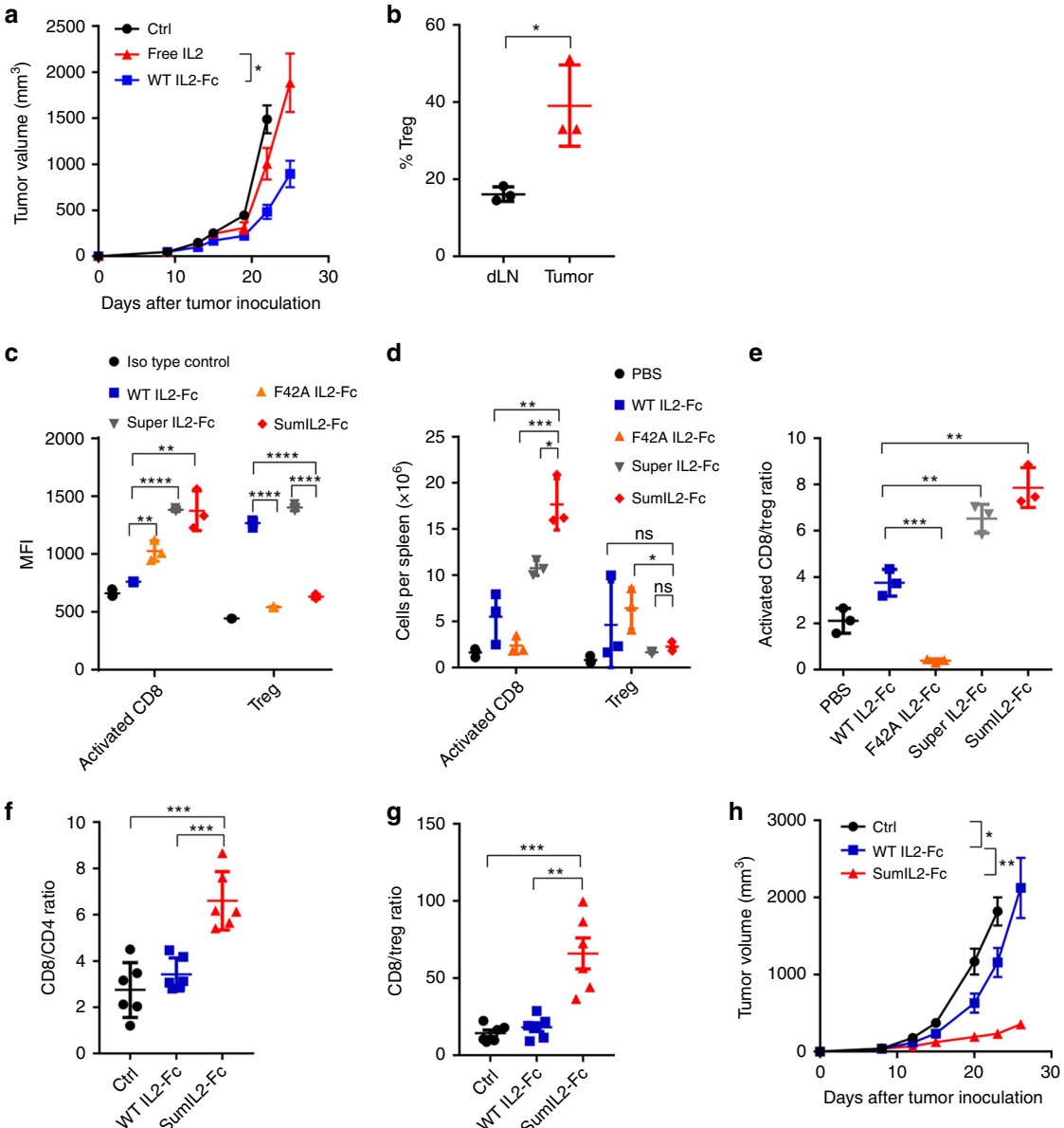

**Fig. 1** IL-2 therapeutic effect is limited by binding to Treg preferentially. **a** WT C57BL/6 mice ($n = 5$/group) were injected subcutaneously with $5 \times 10^5$ B16F10 tumor cells and intratumorally (i.t.) treated with 5 µg of WT IL-2-Fc or same mole number of free IL-2 on day 9, 13, and 15. **b** WT C57BL/6 mice ($n = 3$–5/group) were injected subcutaneously with $5 \times 10^5$ B16F10 cells. The tumor tissues were collected on day 14. The percentages of Treg cells among CD4$^+$ T cells within tumor tissue and draining lymph node were detected by flow cytometry. **c** Foxp3-GFP mice spleen cells were incubated with IL-2 variants-Fc fusion protein, followed by anti-huIgG-PE staining. **d**, **e** WT C57BL/6 mice ($n = 3$–5/group) were injected intraperitoneally (i.p.) daily with either phosphate-buffered saline (PBS, as a control), 2 µg WT IL2-Fc, F42A IL2-Fc, super IL2-Fc, or sumIL2-Fc respectively for 5 days. Total cell counts of host CD3$^+$CD8$^+$CD44$^{high}$ activated T cells, host CD3$^+$CD4$^+$Foxp3$^+$ T cells (**d**) and activated CD8/Treg ratio (**e**) were determined in the spleens by flow cytometry. **f**, **g** WT C57BL/6 mice ($n = 3$–5/group) were injected subcutaneously with $5 \times 10^5$ B16F10 tumor cells and i.t. treated with 5 µg of WT IL-2 or sumIL2-Fc on day 9. Tumor tissues were collected on day 12. CD8/CD4 (**f**) and CD8/Treg (**g**) in tumor tissues were analyzed by flow cytometry. **h** WT C57BL/6 mice ($n = 5$/group) were injected subcutaneously with $5 \times 10^5$ B16F10 and i.t. treated with 5 µg of WT IL-2 or sumIL2-Fc on days 9, 12, and 15. Two way ANOVA tests were used to analyze the tumor growth data. Unpaired T-tests were used to analyze the other data. ns (not significant), *$P < 0.05$, **$P < 0.01$, ***$P < 0.001$, and ****$P < 0.0001$. One of the two or three representative experiments is shown

induced by sumIL2-Fc than that in WT IL2-Fc (Fig. 1e). Furthermore, we injected the same molar of WT IL2-Fc and sumIL2-Fc into B16F10 tumor tissue to compare their respective TIL characteristics (Fig. 1f–h; Supplementary Fig. 3c, d). SumIL-2 significantly enhanced the percentage of CD8$^+$ T cells amongst intratumoral CD3$^+$ T cells, the CD8/CD4 cell ratio and CD8/Treg ratio. In addition, sumIL2-Fc induced a significant higher number of CD8$^+$ T cells in tumor tissue than WT IL2-Fc

(Supplementary Fig. 3c), and did not result in more Tregs (Supplementary Fig. 3d). Most importantly sumIL-2 showed superior antitumor efficacy to WT IL-2. As shown in Fig. 1h, the intratumoral injection of low-dose WT IL2-Fc could not control tumor growth, but the same molar of sumIL-2-Fc effectively inhibited tumor growth. It is therefore demonstrated that sumIL-2 is more effective than WT IL-2. Thus, this sumIL-2 indeed achieved an enhanced stimulation to CD8$^+$ T cells but not Treg cells.

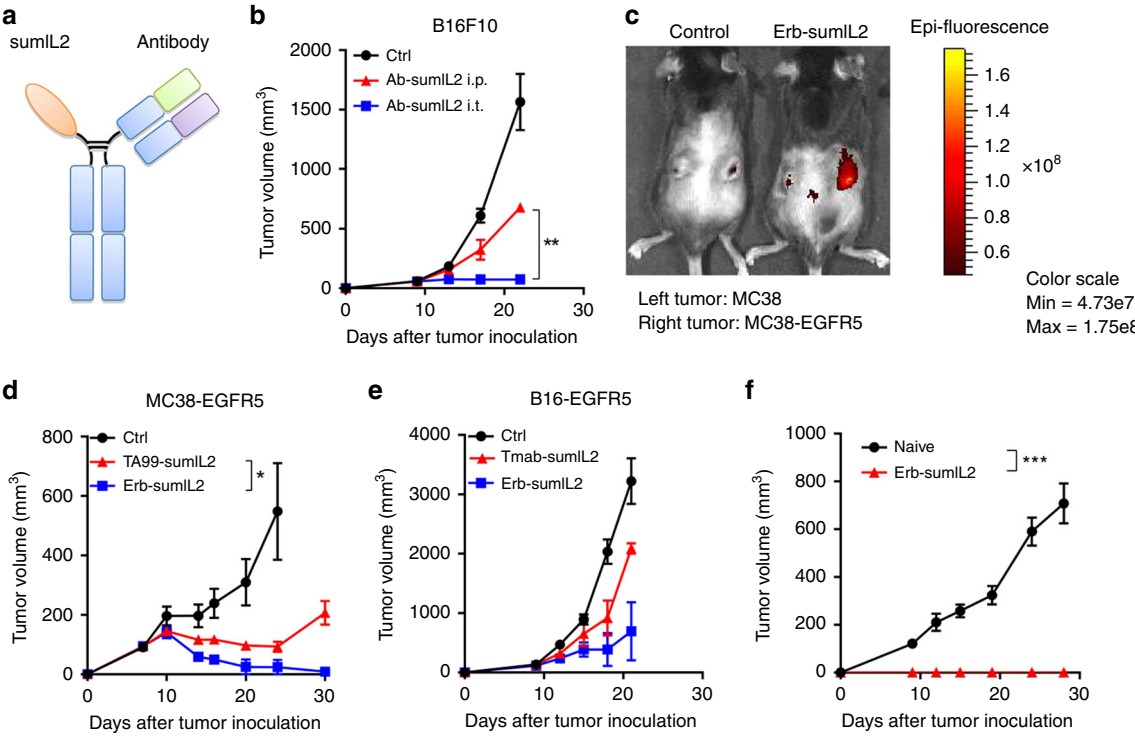

**Fig. 2** Targeting sumIL-2 to tumor site significantly improves its therapeutic effect. **a** The schematic model of Erb-sumIL2. SumIL-2 is fused to the Fc and forms heterodimer with anti-EGFR. **b** C57BL/6 mice ($n = 3–5$/group) were injected subcutaneously with $5 × 10^5$ of B16F10 tumor cells, and 10 μg of Erb-sumIL2 was administered intraperitoneally (i.p.) or i.t. on days 9, 13, and 17. The growth of the tumor was measured and compared twice a week. **c** Fluorescence images of MC38(left) and MC38-EGFR5(right) tumor-bearing mice treated with a single dose of PBS or 25 μg Erb-sumIL2 (i.v.). **d** C57BL/6 mice were ($n = 5$/group) injected subcutaneously with $5 × 10^5$ of MC38-EGFR5 cells, and then i.v. treated on days 7 and 10 with PBS, 25 μg of Erb-sumIL2 or 25 μg TA99-sumIL2. **e** C57BL/6 mice ($n = 3–5$/group) were injected subcutaneously with $5 × 10^5$ B16-EGFR5 cells, and then i.v. treated on days 9, 12, and 15 with PBS, 25 μg of Erb-sumIL2 or 25 μg Tmab-sumIL2 (anti-Her2-sumIL2). **f** WT C57BL/6 mice or mice with tumor clearance by Erb-sumIL2 ($n = 5$/group) were injected subcutaneously with $3 × 10^6$ of MC38-EGFR5, and the growth of the tumor was measured and compared twice a week

**Tumor targeting is critical for the antitumor effect of sumIL-2 therapy**. In order to decrease IL-2 binding to its high-affinity receptor on activated T cells in circulation or other tissues before it reaches the TME, we utilized a heterodimeric sumIL-2 with reduced binding affinity. This sumIL-2 is composed of anti-human EGFR-sumIL-2 (abbreviated as Erb-sumIL2), featuring a sumIL-2-Fc monomer in one arm and anti-human EGFR in another arm (Fig. 2a). It is a next-generation IL-2 fusion protein for efficient delivery of sumIL-2 not only to target EGFR positive tumor tissue but also to target CTL in TME for resulting in more effective tumor killing. A highly purified Erb-sumIL2 heterodimer was characterized via SDS-PAGE and half-life analysis in vivo (Supplementary Fig. 4a and 4b). We found that the half-life of Erb-sumIL2 was about 8 h, much longer than that of the current FDA-approved free IL-2, which generally clears from circulation within minutes[33]. In addition, to test the therapeutic effect of targeted sumIL-2 therapy, we also generated mouse tumor cell lines preferentially expressing mouse EGFR with few site mutations which can be recognized by Cetuximab, the FDA-approved anti-human EGFR antibody for proof of concept of tumor targeting.

We detected whether tumor targeting is essential for sumIL-2 treatment, by comparing the therapeutic efficacy between intratumoral injection and systemic administration of low dose Erb-sumIL2 to B16F10, MC38, and A20 tumor models (Fig. 2b, Supplementary Fig. 4c, d). Local, but not systemic, administration of Ab-sumIL2 could control the tumor growth, suggesting that tumor-targeted sumIL-2 might be necessary for effective antitumor immunity. The potency of systemic delivery of Erb-

sumIL2 targeting to the EGFR positive tumor was tested. We first subcutaneously injected MC38 cells on the left flank of WT C57/BL6 mice and the same number of MC38-EGFR5 cells on the right flank. Next, 25ug of Cy5.5-labeled Erb-sumIL2 was injected intravenously (i.v.) on day 10 post tumor inoculation. Immunofluorescence analysis of tumors showed intense staining in the MC38-EGFR5 tumor (Fig. 2c), suggesting that Erb-sumIL2 specifically targets the EGFR positive tumor. The fluorescence in the MC38-EGFR5 tumor can be still detected on day 3 post-Erb-sumIL2 administration (Supplementary Fig. 4e). To test the therapeutic effect of Erb-sumIL2, Erb-sumIL2 or control fusion protein was administrated i.v. to MC38-EGFR5 tumor-bearing mice (Fig. 2d). We observed that Erb-sumIL2 significantly enhanced the tumor control when compared with the control fusion protein. The same result was obtained from the B16-EGFR5 tumor model (Fig. 2e). In addition, the antitumor effects of the Erb-sumIL2 were better than that of other Erb-IL-2 variants (Supplementary Fig. 4f), indicating that reducing CD25-binding affinity or increasing CD122-binding affinity was not enough to improve the function of IL-2. In addition, Erb-sumIL2 could significantly prolong the survival of mice (Supplementary Fig. 4g). Mice with tumor clearance after Erb-sumIL2 treatment were rechallenged 3 months later by subcutaneous injection of MC38-EGFR5 with five times higher dose than the initial inoculation. None of these rechallenged mice redeveloped tumors while tumors grew progressively in all of the naïve control mice (Fig. 2f). This result indicates that mice with tumor clearance after Erb-sumIL2 treatment had acquired protective memory immunity.

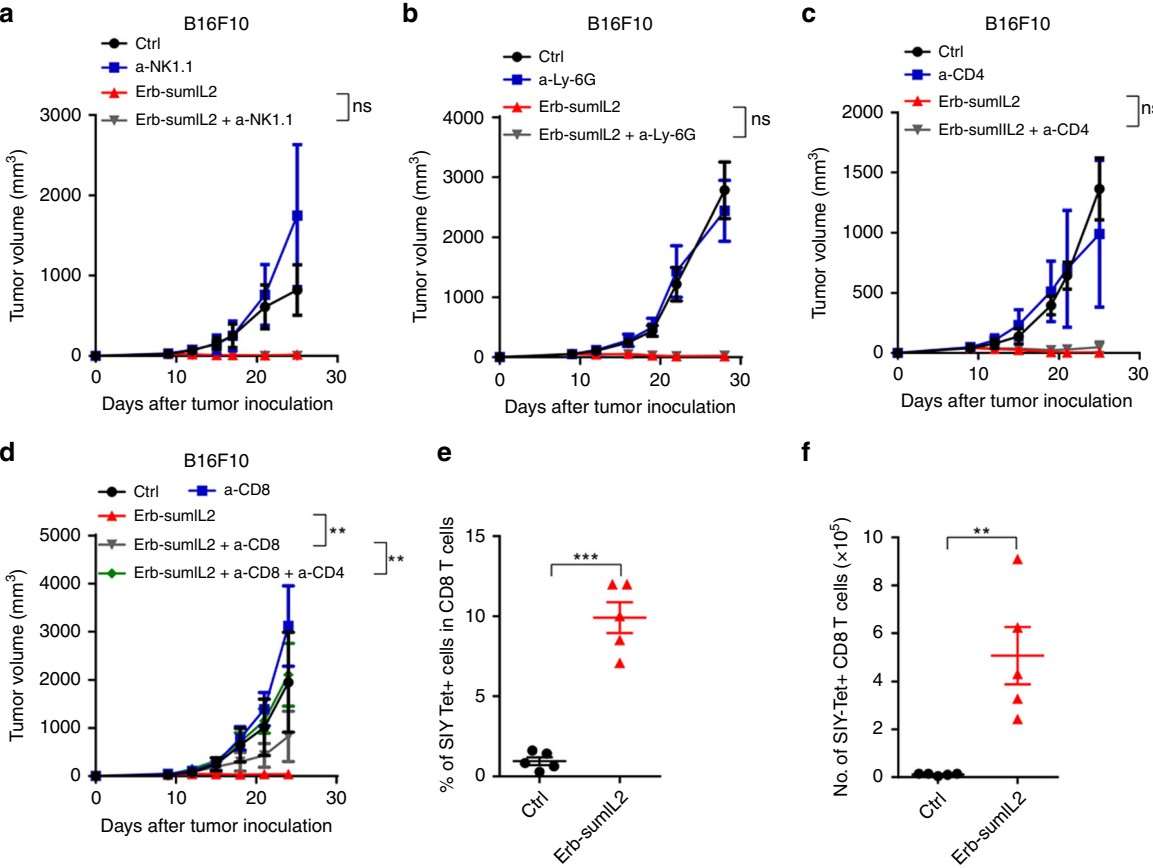

**Fig. 3** CD8+ T cells are essential for Ab-sumIL2 therapy. **a–d** WT C57BL/6 mice (n = 5/group) were injected subcutaneously with $5 \times 10^5$ B16F10 cells, and i.t. treated with 10 μg of Erb-sumIL2 on days 9, 12, and 15. Anti-NK1.1 (**a**) and anti-Ly-6G (**b**) were injected at a dose of 400 μg per antibody, 8 days after tumor inoculation and every 4 days thereafter for 3 times. Anti-CD4 (**c**) and anti-CD8 (**d**) were injected at a dose of 200 μg on the same day as Erb-sumIL2. **e**, **f** WT C57BL/6 mice (n = 5/group) were injected subcutaneously with $7.5 \times 10^5$ B16-SIY cells and i.t. treated with 10 μg of Erb-sumIL2 on day 9. The tumor tissues were collected on day 15. Antigen-specific CD8 T cells in the tumor site were detected using SIY tetramer. The frequency of SIY-specific CD8 T cells among total CD8 T cells (**e**) and the number of SIY-specific CD8 T cells (**f**) were analyzed by flow cytometry. Two way ANOVA tests were used to analyze the tumor growth data. Unpaired T-tests were used to analyze the other data. ns (not significant), $*P < 0.05$, $**P < 0.01$, $***P < 0.001$, and $****P < 0.0001$. One of the two or three representative experiments is shown

**CD8+ T cells but not NK cells are necessary for the efficacy of Ab-sumIL2 tumor therapy.** The B16 tumor is sensitive to NK cells and CTLs[34,35], both of which can be activated by IL-2. However, the major functional cells for Ab-sumIL2 induced antitumor response are not known yet. Here, we evaluated the essential role of innate and adaptive immunity during sumIL-2 treatment; B16F10-bearing WT C57BL/6 mice and Rag1 KO mice were treated with 10 μg of Erb-sumIL2. Tumor growth in WT mice was inhibited by Erb-sumIL2, but the therapeutic effect of Erb-sumIL2 was abolished in Rag1 KO mice despite a high percentage of NK cells, revealing that therapeutic effect of Erb-sumIL2 requires adaptive immunity (Supplementary Fig. 5a). We further tested whether innate immune-associated cells are essential for Erb-sumIL2 treatment by depleting NK cells or neutrophil cells during Erb-sumIL2 treatment in B16F10-bearing WT mice. Ultimately, neither the depletion of NK cells (Fig. 3a) nor neutrophil cells affected the therapeutic effects of Erb-sumIL2 (Fig. 3b). We further investigate which subsets of T cells are critical for Erb-sumIL2 treatment; CD4-depleting antibody was administered during Erb-sumIL2 treatment and the therapeutic effects of Erb-sumIL2 were not eliminated (Fig. 3c). Finally, we used anti-CD8 Ab to deplete CD8 T cells during Erb-sumIL2 treatment, the depletion of CD8+ T cells greatly abolished the therapeutic effects, but Erb-

sumIL2 could still partially control tumor growth (Fig. 3d). This data suggested that other cell types play an essential role in the absence of CD8 T cells. When we depleted both CD4 and CD8 T cells during Erb-sumIL2 treatment, the therapeutic effects were abolished entirely. These data further support the conclusion that the therapeutic effect of Erb-sumIL2 is dependent on T cells but not NK cells.

To test which subsets of T cells are expanded, we first analyzed the B16F10-infiltrating T cells using flow cytometry (Supplementary Fig. 5b). The ratios of intratumoral CD8/CD4, CD8/Treg, and CD4/Treg significantly increased after Erb-sumIL2 treatment; similar results were obtained in the MC38 tumor model (Supplementary Fig. 5c). We further evaluated the tumor antigen-specific CD8+ T cell response. SIY (SIYRYYGL) is a synthetic peptide associated with the class I molecule $K^b$ which can be recognized by 2C TCR transgenic T cells. In B16-SIY model, SIY as an antigen can be presented by SIY expressing tumor cell or cross-presented by antigen-presenting cells (APCs) expressing $K^b$. We use associated tetramer to detect SIY-specific CD8 T cells. In order to evaluate whether and how Ab-sumIL2 regulates antigen-specific T cells inside the tumor, we continued to use an intratumoral injection of Erb-sumIL2. B16-SIY tumor cells were subcutaneously injected to WT C57BL/6 mice and tumor antigen-specific CD8+ T cells were detected via Flow cytometry

by SIY-specific tetramer staining, 6 days after the initial Erb-sumIL2 treatment (Supplementary Fig. 5d; Fig. 3d). The percentage and the absolute number of SIY-specific CD8[+] T cells inside the tumor increased on day 6 compared with the control group, suggesting that Erb-sumIL2 can induce a tumor-specific T cell response. We also used the MC38-OVA mouse tumor model to confirm that Erb-sumIL2 can induce a large number of tumor antigen-specific CD8 T cells (Supplementary Fig. 5e). It has been shown that a large amount of tumor-specific T cells inside tumor tissues are dysfunctional[36]. We propose that lack of IL-2 limits the proper expansion of antigen-specific T cells (SIY-reactive T cells in the B16-SIY model and OVA-reactive T cells in MC38-OVA model), and pre-existing antigen-reactive T cells can be reactivated and expanded when IL-2 is sufficiently provided. Here, we used i.t. injection of Erb-sumIL2 into EGFR negative B16-SIY and MC38-OVA tumor to assess the change of antigen-specific T cell inside the tumor, but not for the targeting delivery of sumIL-2. One major side-effect of IL-2 treatment is vascular leak syndrome (VLS). We observed that while large numbers of CTLs were induced by Erb-sumIL2 treatment and this did not cause pulmonary edema (Supplementary Fig. 6), suggesting that Erb-sumIL2 treatment does not cause serious systemic toxicity at the therapeutic range. Together, the data suggest that Erb-sumIL2 can reduce toxicity while enhancing antitumor efficacy through preferentially activating CTL.

**Pre-existing intratumoral CTLs are essential and sufficient for Ab-sumIL2 therapy, and activated CTLs can control a distal tumor.** IL-2 treatment can activate T cells in not only tumor tissues but also other tissues and the DLNs. With this in mind, we explored whether TILs or T cells from outside tumor tissues, especially DLN, were necessary and or sufficient for Erb-sumIL2 treatment. FTY720 is a structural analogue of the sphingosine-1-phosphate and is derived from myriocin. It can potently inhibit the egression of T cells from the LN into the circulation and peripheral tissues. Mice were treated with FTY720 1 day after B16F10 tumor inoculation to diminish peripheral T cells trafficking into tumor tissues (Fig. 4a). After tumors were established, mice were treated with Erb-sumIL2. The antitumor effects of Erb-sumIL2 were completely abolished in the presence of FTY720, confirming that tumor-infiltrating T cells are, in fact, necessary for tumor control by tumor-targeting mutant IL-2 therapy.

We then determined whether the pre-existing CD8[+] T cells were sufficient for Erb-sumIL2 therapy. B16F10-bearing mice were given FTY720 during Erb-sumIL2 treatment and antitumor effects were still well maintained (Fig. 4b), implying that pre-existing T cells are sufficient for mutant IL-2 therapy. Furthermore, when CD8 T cells were depleted during FTY720 and Erb-sumIL2 treatment, the therapeutic effects of Erb-sumIL2 were abrogated and the same results were observed from the MC38 tumor model (Fig. 4c). These data demonstrate that tumor-infiltrating CD8[+] T cells play a necessary and sufficient role in the Erb-sumIL2 treatment. In addition, we found that the frequency of PD-1[+] CXCR5[+] CD8[+] T cells was increased after Erb-sumIL2 treatment. This subset of CD8[+] T cells has been reported to have a potential cytotoxic function in chronic virus infection and cancer[37,38]. It suggests that PD-1[+] CXCR5[+] CD8[+] T cells may play an essential role in Erb-sumIL2 induced antitumor effects. To test whether increased CTLs from TME can have a systemic impact, we then evaluated whether CTLs induced within tumor tissue could circulate to and control a distal tumor. We i.t. treated the local tumor with Erb-sumIL2 and measured the volume of the distal tumor; the growth of the distal tumor was also inhibited (Fig. 4e). It suggests that tumor-infiltrating CD8[+]

T cells are not only sufficient for local-tumor control but also for distal-tumor control.

**SumIL-2 synergizes with targeted therapy to control cold tumors.** The second and third generation of EGFR-TKI therapy can induce a high response rate in patients with EGFR or Her2 dependent cancers, but high relapse rate becomes the major clinical problem. To test whether Ab-sumIL2 can limit the relapse in combination with TKI, we evaluated the therapeutic effects of Erb-sumIL2 on the TUBO, a HER2/*neu*-dependent mammary carcinoma derived from BALB/c mice transgenic with the *neu* oncogene[39]. The *neu* gene is highly expressed on this tumor cell line. First, we detected the therapeutic effects of sumIL-2 on TUBO tumor model. Unlike the B16F10 and MC38 tumor models, Erb-sumIL2 could not control tumor growth of established TUBO with three intratumoral administrations of 10 μg of Erb-sumIL2 in this murine model (Fig. 5a). We wondered if the failure to respond to sumIL-2 therapy in the TUBO model was due to insufficient TILs for established tumors, similar to clinical EGFR/HER2-driven tumors. We measured and compared the frequency of T cells inside the TUBO tumor. Indeed, many more TILs were found in the MC38 and B16F10 tumors than in the TUBO model. In addition, the percentage of CD8[+] T cells amongst CD3[+] T cells is higher in the MC38 and B16F10 tumors than that of the TUBO tumor (Fig. 5b). Low numbers of intra-tumoral CTLs are commonly observed in the majority of patients with TKI resistant cancer[40]. Most of those patients also fail to respond to immunotherapy[41]. We proposed that TKI could induce more TILs in tumor and Ab-sumIL2 may synergize with targeted therapy to kill a large number of tumor cells and overcome resistance to either single treatment for better controlling cold tumors.

In the clinic, TKI has been recommended as the first-line treatment for those oncogen-dependent tumors, but tumor relapse is often observed several months after treatment. In our study, we first treated TUBO-bearing BALB/c mice with afatinib, a second-generation TKI agent (Fig. 5c–e). After a single treatment of afatinib, the ratio of CD3[+] T cells/tumor cells in the tumor was indeed increased (Fig. 5c) and the tumor was initially controlled very well but later all the treated tumors relapsed (Fig. 5e). Erb-sumIL2 did not make any change for tumor control when used after afatinib treatment. However, the administration of afatinib and the Erb-sumIL2 at the same time effectively control tumor growth and prevent tumor development after being rechallenged with a high dose of tumor cells (Fig. 5f). Altogether, these data suggest that proper use of Ab-sumIL2 could combine with TKI for treating cold tumors. Clinically, immunotherapy is usually applied as a second-line drug. Interestingly, we found sumIL-2 therapy does not control relapsed tumors after TKI treatment (Fig. 5d). Thus, the data suggest that IL-2 should become the first-line treatment with TKI for advanced EGFR/HER2 dependent tumors.

**Preoperative sumIL-2 therapy can eradicate metastatic disease.** Tumor surgery is the most effective single modality for cancer patients. However, many patients will relapse due to metastases. Thus, other treatments such as chemotherapy and immunotherapy are used as adjuvant therapies. IL-2 as an immune modulator, which might be another effective therapy to enhance antitumor response. Given using the sumIL-2 at the right time is critical to treatment outcomes, we wondered whether sumIL-2 treatment after surgical removal of tumor burden could induce a long-lasting effect using a standard adjuvant therapy schedule. The 4T1 mammary carcinoma is a good mouse metastatic tumor model, which can spontaneously metastasize from the primary tumor in the

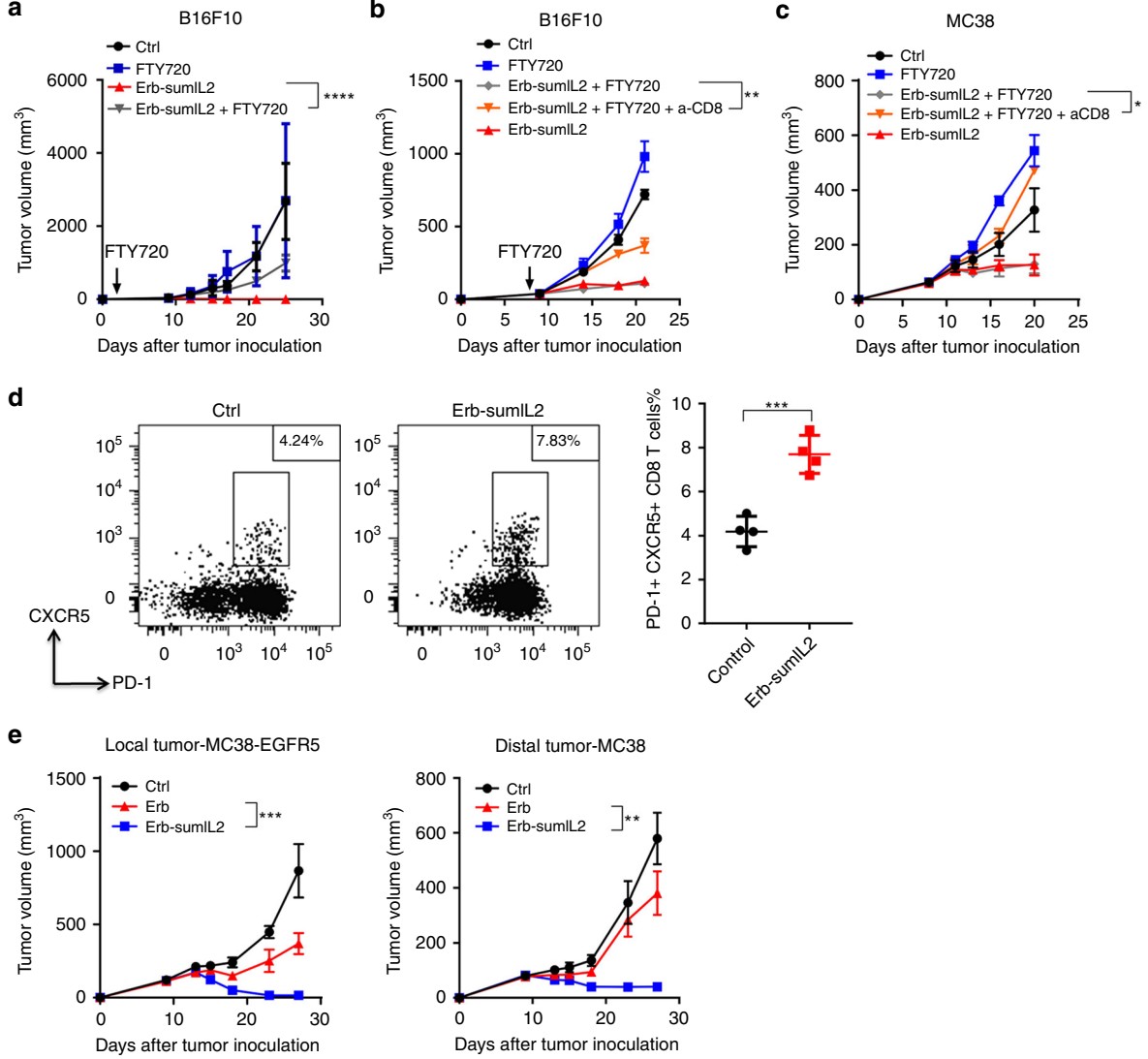

**Fig. 4** Pre-existing intratumoral CTLs are sufficient and essential for Ab-sumIL2 therapy and activated CTLs can control distal tumors. **a** WT C57BL/6 mice ($n = 5$ group) were injected subcutaneously with $5 \times 10^5$ of B16F10 cells, and then i.t. treated on days 9, 12, and 15 with PBS or 10 μg of Erb-sumIL2. For FTY720 blockade, 10 μg FTY720 was given from day 1 and every other day thereafter for 4 times. **b** WT C57BL/6 mice ($n = 5$ group) were injected subcutaneously with $5 \times 10^5$ of B16F10 cells and i.t. treated with 10 μg of Erb-sumIL2 on days 9, 12, and 15. FTY720 blockade and CD8 T cell depletion started on day 8 posttumor inoculation. **c** WT C57BL/6 mice ($n = 5$ group) were injected subcutaneously with $5 \times 10^5$ MC38 cells and i.t. treated with 10 μg Erb-sumIL2 on days 8, 11, and 13. FTY720 blockade and CD8 depletion were performed starting from day 7 of tumor inoculation. **d** WT C57BL/6 mice ($n = 3$–5 group) were injected subcutaneously with $5 \times 10^5$ of B16F10 cells and i.t. treated with 10 μg of Erb-sumIL2 on day 9. Tumor tissues were collected on day 12 and the frequency of PD-1$^+$ CXCR5$^+$ CD8$^+$ T cells was detected by flow cytometry. **e** WT C57BL/6 mice ($n = 5$ group) were injected subcutaneously with $5 \times 10^5$ MC38 (left flank) and MC38-EGFR5 (right flank) cells. MC38-EGFR5 tumor was i.t. treated with 10 μg Cetuximab (Erb) or 10 μg Erb-sumIL2 on days 9, 12, and 15. The volume of both tumors was measured twice a week. Two way ANOVA tests were used to analyze the tumor growth data. Unpaired T-tests were used to analyze the other data. ns (not significant), *$P < 0.05$, **$P < 0.01$, ***$P < 0.001$, and ****$P < 0.0001$. One of the two or three representative experiments is shown

mammary gland to multiple distant tissues. Thus, we used this tumor model to determine whether sumIL-2 could induce an effective antitumor response in eradicating metastatic tumors after the local tumor is surgically removed. Unexpectedly, only those mice in the preoperative sumIL-2 treatment group survived in a much longer-term compared with those that received surgery only or with an adjuvant sumIL-2 treatment group (Fig. 6a). Furthermore, we examined whether T cell responses were involved in the durable therapeutic effects induced by preoperative sumIL-2 treatment. In the 4T1 model, the efficacy of preoperative sumIL-2 treatment was dependent on CD4 and CD8 T cells (Fig. 6b). These data suggest that T cell response induced by preoperative sumIL-2

treatment before tumorectomy can significantly improve the survival of patients with metastatic cancer.

**PD-L1 blockade synergizes with sumIL-2 therapy to control advanced tumors.** Given its impressive effect on enhancing tumor-specific T cell responses, sum-IL-2 may induce the expression of PD-L1 on a tumor cell and/or APCs[42]. This might contribute to tumor relapse after initial response to Ab-muIL2. To detect tumor PD-L1 expression, tumor tissues were collected three days after sumIL-2 treatment. Tumor cells (CD45$^-$), DCs (CD45$^+$ CD11c$^+$ MHCII$^+$), and macrophages (CD45$^+$ CD11b$^+$ F4/80$^+$) were

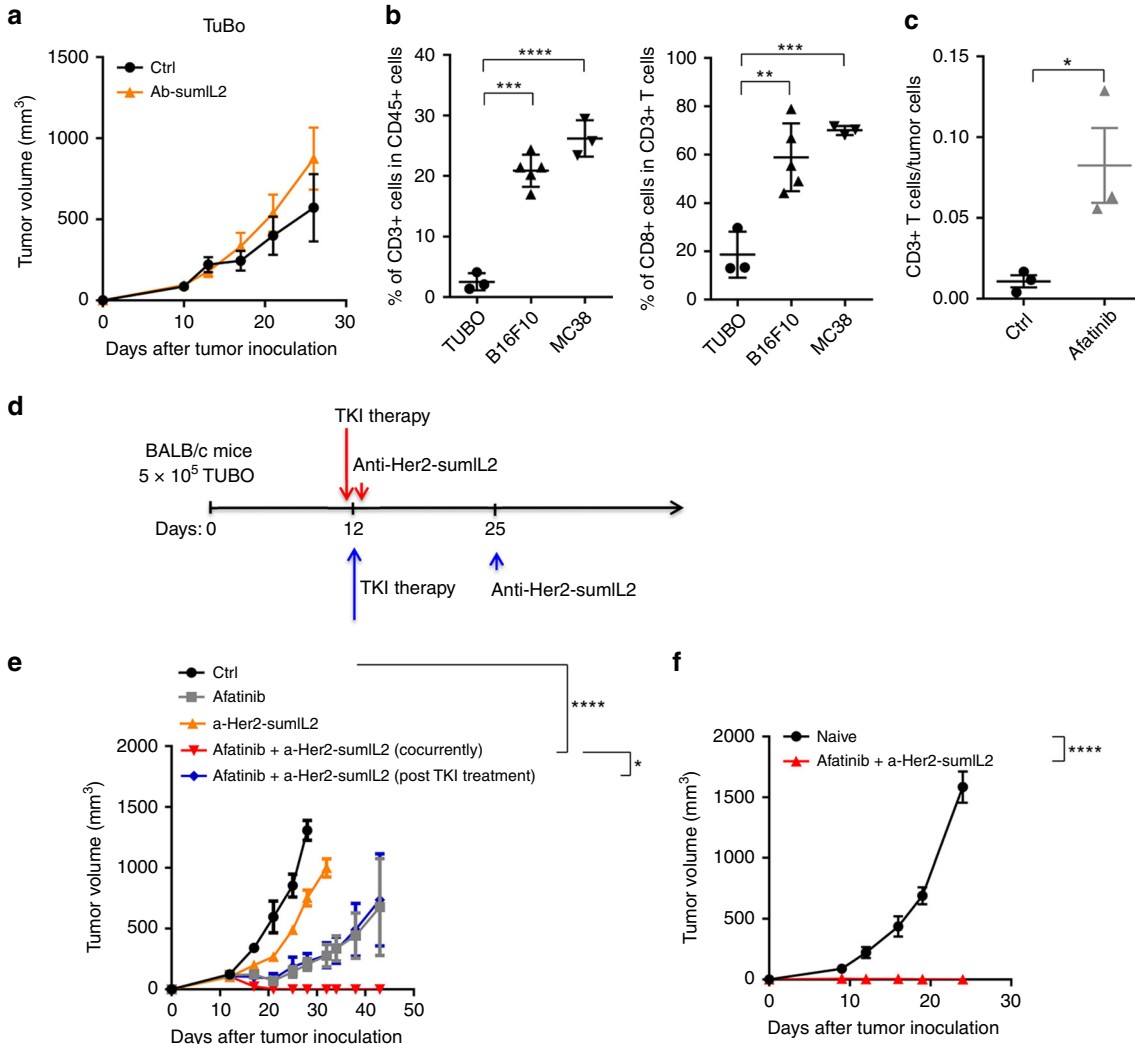

**Fig. 5** First-line TKI therapy can synergize with sumIL-2 therapy to control cold tumor. **a** BALB/c mice ($n = 5$ group) were inoculated subcutaneously with $5 \times 10^5$ of TUBO cells and then were i.v. treated with PBS or 10 μg of Erb-sumIL2 on days 10, 13, and 17. **b** Indicated tumor tissues were collected on day 9 posttumor inoculation. The percentages of CD3$^+$ among CD45$^+$ cells (left) and CD8$^+$ among CD3$^+$ cells (right) were analyzed by flow cytometry. **c** BALB/c mice ($n = 3$–5 group) were inoculated subcutaneously with $5 \times 10^5$ of TUBO cells and then were i.p. treated with PBS or 1 mg of afatinib. TUBO tumor tissues were collected on day 6 post afatinib single treatment. The percentage of CD8$^+$ among CD3$^+$ cells was analyzed by flow cytometry. **d**, **e** BALB/c mice ($n = 5$ group) were inoculated subcutaneously with $5 \times 10^5$ of TUBO cells and then were i.p. treated with PBS or 20 μg of anti-Her2-sumIL2 on either days 12, 15, and 18 or days 25, 28, and 31. For TKI therapy, mice were treated orally with 1 mg of afatinib on days 12 and 17. The tumor growth was measured. **f** WT BALB/c mice or mice with tumor clearance by afatinib and anti-Her2-sumIL2 were injected subcutaneously with $3 \times 10^6$ TUBO cells. The growth of tumor was measured and compared twice a week. Two way ANOVA tests were used to analyze the tumor growth data. Unpaired T-tests were used to analyze the other data. ns (not significant), *$P < 0.05$, **$P < 0.01$, ***$P < 0.001$, and ****$P < 0.0001$. One of the two or three representative experiments is shown

analyzed. Compared with the tumors of the control group, the upregulation of PD-L1 was observed on DC cells in tumor tissues (Fig. 7a). This data suggest that PD-1/PD-L1 blockade may enhance the antitumor effects of Ab-sumIL2, vice versa.

The immune checkpoint blockades have been reported broadly for their antitumor effects. PD-1/PD-L1 signaling inhibits effector T cell response in the tissue and tumor. Blocking PD-1/PD-L1 could derepress the pre-existing immune response. Nonetheless, limited numbers of reactivated T cells may not be enough for tumor control. IL-2 is an inducer for T cell proliferation, so it can further amplify pre-existing T cell responses. Following this rationale, we detected whether sumIL-2 therapy could improve the function of the checkpoint blockade (Fig. 7b). Anti-PD-L1 single treatment can partially control tumor growth, but additional sumIL-2 significantly enhances the antitumor effects

of checkpoint blockade alone. These data thus demonstrate that sumIL-2 can overcome resistance to checkpoint blockade.

## Discussion

While IL-2, as a growth factor for T cells, was initially shown to be a promising and effective antitumor therapy for human cancers, the broader application of this cytokine is limited due to its extremely short half-life, severe toxicity, lack of tumor targeting, and preferential activation of Treg. To overcome these defects, we designed a next-generation IL-2 comprising a tumor-targeting antibody and a mutated human IL-2 that exhibits reduced CD25 binding and enhanced CD122 binding. This new sumIL-2 immunocytokine has a longer half-life, efficient tumor targeting, less toxicity, and preferential binding to CTLs but not Tregs.

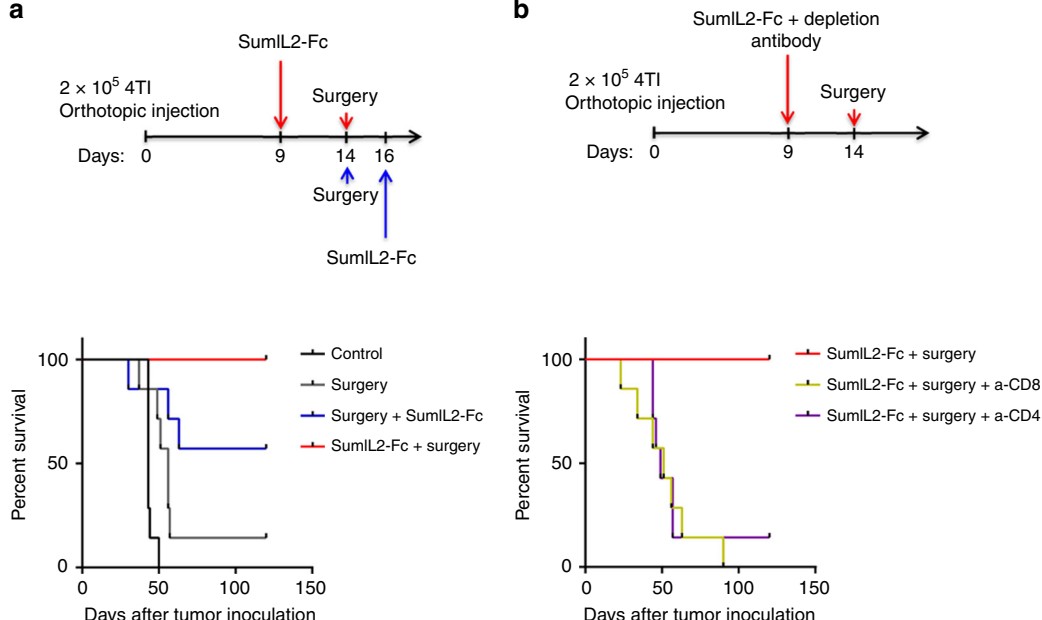

**Fig. 6** Preoperative therapy with sumIL-2 enhances efficacy in eradicating metastatic disease. BALB/c WT mice ($n = 7$/group) were injected with $2 \times 10^5$ 4T1 mammary carcinoma cells into the mammary fat pad. **a** As indicated in the schematics, one group of mice were i.p. treated with 10 μg sumIL-2-Fc on days 9, 11, and 13 before surgery; one group of mice were treated on days 16, 18, and 20 after surgery. All primary tumors were resected on day 14, except for the no-surgery group which as a control. **b** Groups of mice ($n = 7$/group) were treated with preoperative 10 μg sumIL-2-Fc on days 9, 11, and 13. All primary tumors were resected on day 14. For depletion of CD4$^+$ or CD8$^+$ T cells, 200 μg anti-CD4 or anti-CD8 depletion antibody was injected on the same day of sumIL-2-Fc treatment

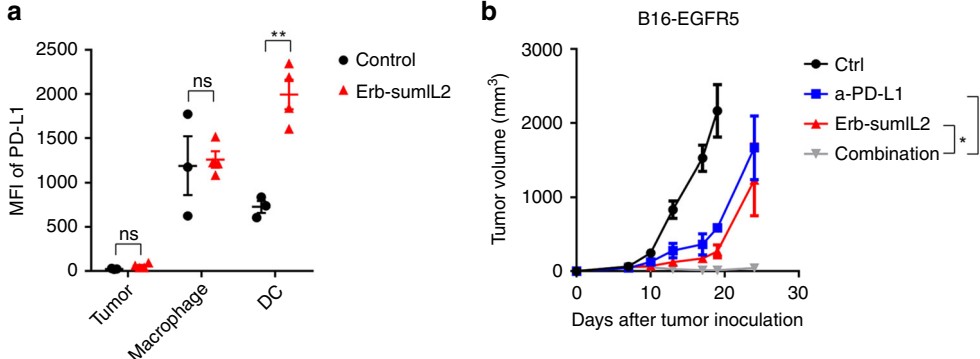

**Fig. 7** SumIL-2 therapy overcomes checkpoint blockade resistance. **a** C57BL/6 mice ($n = 3$–5/group) were inoculated with $5 \times 10^5$ B16F10 cells and treated intravenously with 10 μg of Erb-sumIL2 on day 9. Three days after treatment, tumors were removed and digested into single-cell suspensions. PD-L1 expression on myeloid cells and tumor cells was detected by flow cytometry. **b** WT C57BL/6 mice ($n = 5$/group) were injected subcutaneously with $5 \times 10^5$ of B16F10-EGFR5 cells and i.p. treated with 25 μg of Erb-sumIL2 or/and i.t. treated with 50 μg of anti-PD-L1 on days 8, 11, and 14. Two way ANOVA tests were used to analyze the tumor growth data. Unpaired T-tests were used to analyze the other data. ns (not significant), *$P < 0.05$, **$P < 0.01$, ***$P < 0.001$, and ****$P < 0.0001$. One of the two or three representative experiments is shown

Several IL-2 variants have been reported to modify their binding to specific subsets of T cells. Dimeric IL-2Rs are expressed at high levels on Treg and antigen-experienced memory CD8$^+$ T cells[12]. F42A mutant IL-2 with lower affinity to IL-2Rα could reduce its binding potential to Treg but may also limit its bind to CTL. Super IL-2 with increased CD122 binding could stimulate more CD8$^+$ T cells. Although super IL-2 can induce pSTAT5 activation of cells lacking IL-2R alpha[15], this mutation might also improve its binding to the heterotrimeric receptor on Treg cells. Therefore, we think it is necessary to generate a new generation IL-2 that reduces binding to CD25 while increasing binding to CD122 for an improved CTL/Treg ratio. Indeed, sumIL-2 achieved an enhanced activation of CD8$^+$ T cells but not Treg cells, and showed superior antitumor efficacy to WT IL-2.

This sumIL-2 has shown not to induce harmful side-effects like VLS. CD25 on pulmonary endothelial cells was suggested to be a major contributor to VLS that was induced by recombinant human IL2[43]. Thus, reducing CD25 binding could lead to less toxicity.

Although polyethylene glycol (PEG) conjugations on the particular position of IL-2 has been reported to reduce its affinity to IL-2Rαβγ[44,45], quantity production and precision quality control are hard to achieve for the clinical use of this PEG-modified IL-2. The lack of tumor targeting for PEG-IL-2 might also limit its activity on TILs. Since a sustained presence of IL-2 in tumors is critical for producing therapeutic effects[23,44], we constructed a tumor-targeting Erb-sumIL2 fusion protein, which can prolong the IL-2 half-life by avoiding renal filtration of the low molecular

weight protein and by delivering sumIL-2 to the tumor tissue to activate tumor-infiltrated T cells. In order to achieve effective targeting, the Erb-sumIL2 was constructed as a heterodimer. The monomer structure of sumIL-2 on Erb-sumIL2 may reduce its affinity to IL-2R expressing cells and its consumption by peripheral cells.

Numerous current cancer immunotherapies are focused on immune checkpoint blockade. While releasing brakes on dysfunctional CTLs is likely to improve TIL readiness, additional growth factors for CTLs may be required to optimize their proliferation and activation. It has been reported that IL-2 signaling is essential for optimal primary and secondary CD8[+] T cell responses[46–48]. However, sequencing data from online databases show that IL-2 is not expressed or expressed in extremely low concentrations in tumor tissue. This alludes to the possibility that IL-2 could be a right candidate cytokine for T cell activation and tumor killing. Ab-sumIL2 can efficiently deliver sumIL-2 to tumors, promote the activation and proliferation of intratumoral antigen-specific CD8[+] T cells, and further increase the ratio of CD8/Treg cells. Thus, the significantly enhanced antigen-specific CD8[+] T cells induced by Ab-sumIL2 could achieve tumor regression and long-term memory. Consistently, our results show that pre-existing intratumoral CD8[+] T cells are necessary and sufficient for effective tumor therapy by Ab-sumIL2. Our data demonstrate that tumor-targeting Ab-sumIL2 is a breakthrough for immunocytokines for cancer therapy.

Higher numbers of TILs are associated with better clinical outcomes in many tumors[49,50]. TILs are also important biomarkers for predicting responses to cancer therapy. In our study, we found that TILs were associated with the effectiveness of sumIL-2 therapy. TUBO is a Her2-dependent tumor model with low TILs (cold tumor). In the clinic, TKI is the first-line therapy for treating EGFR mutant positive cancer since most of the patients have fewer TILs and do not respond to immunotherapy. After standard TKI therapy, most patients have initial responses including impressive complete responses but undergo relapse or develop metastasis. Consequently, creating an appropriate regimen for tumor therapy that can reduce tumor burden while increase TILs is of utmost importance. We propose that besides direct tumor killing, TKI therapy may have two advantages for combination therapy. TKI treatment can not only reduce tumor burden, but tumor immunogenicity can also be enhanced after TKI therapy; both outcomes may set a platform and promote the function of immunotherapy. Conversely, immunotherapy could further promote the therapeutic effects of TKI by facilitating tumor killing, including TKI resistant clones, thus overcoming resistance. We found that the ratio of CD3[+] T cell/tumor cells indeed increased after TKI therapy. Nonetheless, tumor-infiltrating T cells are not sufficient in controlling tumors and result in tumor relapse. Appropriate timing of IL-2 can rapidly activate and expand T cells to coordinate well with TKI therapy.

Surgery may be the most exploited modality to reduce tumor mass. However, the antitumor immune response is necessary for killing residual tumors or metastatic tumors. In our study, we observed that sumIL-2 was unable to induce an effective antitumor response for eradicating metastatic tumors when applied after the local tumor was surgically removed. In contrast, preoperative sumIL-2 treatment can significantly extend the survival of mice. Further experiments confirmed that this effect was T cell-dependent. These data suggest that preoperative sumIL-2 may prime an effective antitumor immunity in peripheral lymphoid organs and primary tumor tissues, which can eradicate residual tumors. Therefore, our study also provides a rational immunotherapy and surgery combination approach.

Employing antibodies to block the immune checkpoint PD-1/PD-L1 has been a successful approach in immunotherapy, but the complete response rate is still fairly low. PD-L1 blocking antibodies to releases the "immune brake" on activated T cells, but this might not be adequate to expand CTLs rapidly. However, IL-2 can activate CTLs to produce IFNs that might induce PD-L1 in the TME to limit further CTL function. Therefore, it is logical to combine both IL-2 and PD-1 blockade to synergize their effect on immune responses for tumor control, especially during TKI treatment.

Overall, our study revealed the therapeutic potential of tumor-targeting sumIL-2, a next-generation IL-2, for not only increasing half-life and reducing toxicity but also enhancing treatment effect by efficiently targeting CD8[+] T cells inside tumor tissues. SumIL-2 can overcome the resistance to TKI treatment and TILs are critical for the responsiveness to sumIL-2 therapy. Preoperative sumIL-2 treatment could induce robust antitumor immune responses for eradicating the metastatic disease. Furthermore, sumIL-2 can function as an "immune accelerator" to enhance the therapeutic effects of checkpoint blockades. Altogether, our study has provided a novel, combinatory therapeutic strategy for effectively treating patients with EGFR-driven cancer.

## Methods

**Mice**. WT C57BL/6NCrl and BALB/c mice were purchased from the Vital River Laboratories, and C57BL/6 Rag1 KO mice were purchased from the model animal research center of Nanjing university. Foxp3-GFP mice (BALB/c background) were kindly provided by Dr Yong Zhao at the Institute of Zoology, CAS. All mice were maintained under specific pathogen-free conditions in the animal facility of the Institute of Biophysics. Animal care and experiments were carried out under institutional protocol and guidelines. All studies were approved by the Animal Care and Use Committee of the Institute of Biophysics.

**Cell lines and reagents**. B16F10, MC38, A20, and 4T1 cells were purchased from ATCC. TUBO was cloned from a spontaneous mammary tumor in a BALB/c Neu-transgenic mouse[39]. B16-EGFR5 and MC38-EGFR5 were selected from single-cell clones after being transduced by lentivirus-expressing the mutant mouse EGFR which can bind to cetuximab. TUBO, MC38, B16F10, and the derivatives were cultured in 5% $CO_2$ and maintained in vitro in Dulbecco's modified Eagle's medium, supplemented with 10% heat-inactivated fetal bovine serum, 2 mmol/l L-glutamine, 0.1 mmol/l Minimum Essential Medium nonessential amino acids, 100 U/ml penicillin, and 100 mg/ml streptomycin. 4T1 and A20 cells were maintained in vitro in RPMI 1640 medium. All cell lines were routinely tested using mycoplasma contamination kit (R&D). Anti-CD8 (TIB210), FcγRII/III blocking Ab (2.4G2), anti-CD4 (GK1.5), and anti-NK1.1 (PK136) Abs were produced in-house. Anti-PD-L1 (10 F.9G2) and anti-Ly-6G (1A8) Abs were purchased from Bio X Cell (USA). The TKI-Afatinib was purchased from Shanghai Bojing Chemical Co., Ltd FTY720 was purchased from Sigma.

**Tumor inoculation and treatments**. Approximately $5–7.5 \times 10^5$ of MC38, MC38-EGFR5, B16F10, B16-EGFR5, B16-SIY, or TUBO cells were injected subcutaneously into the right flank of 6–8-week-old mice. $2 \times 10^5$ of 4T1 cells were injected into mammary fat pad. For the double-tumor model, $5 \times 10^5$ of MC38 and MC38 were injected subcutaneously into the left and right flank of 6–8-week-old C57BL/6 mice. Tumor volumes were measured twice a week and calculated (length × width × height/2). After the tumor was established, mice were treated with three intratumoral injections of 10 μg of Erb-muIL2 or control Ab-sumIL2, every three days for three times. There were two controls for Ab-sumIL2 in our study. One was an anti-human Her2-super mutant IL-2 (Tmab-sumIL2) and the other was an anti-mouse TRP1-super mutant IL-2 (TA99-sumIL2). For systemic injection of Ab-sumIL2, 25 μg of Ab-sumIL2 was injected intravenously or intra-peritoneally into tumor-bearing mice twice. To block lymphocyte trafficking, mice were injected intravenously with 20 μg of FTY720 on day 1 posttumor inoculation. A total of 10 μg of FTY720 was administered every other day for three times to maintain the blockade. In some experiments, FTY720 was given 1 day before Ab-sumIL2 treatment. For the 4T1 tumor model, the primary tumor was surgically removed, 10 μg of sumIL2-Fc was given pre or post surgery. For TKI treatment, tumor-bearing mice were treated with Afatinib (TUBO 50 mg/kg) every 4 days for a total of two or three doses through gavage. For depletion of different types of cells, a 200 μg dose anti-CD8 (clone TIB210) or anti-CD4 (GK1.5) was injected i.p. 1 day before Ab-sumIL2 treatment. Anti-NK1.1 (clone PK136) and anti-Ly-6G (clone 1A8) were injected i.p. at a dose of 400 μg per antibody, 1 day before Ab-sumIL2 and every 4 days thereafter.

**Flow cytometric analysis**. For analysis of infiltrating cell number and the ratio of CD8/Treg, tumor-bearing mice were sacrificed and tumors were harvested 3 days after a single treatment. For analysis of tumor-specific CD8[+] T cells, mice were killed, and tumors were harvested 3 and 6 days after a single treatment. Single-cell

suspensions were prepared and incubated with anti-CD16/32 (anti-FcγIII/II receptor, clone 2.4G2) for 30 min and then stained with conjugated Abs against CD45 (30-F11), CD3Ɛ (clone 145-2C11), CD4 (RM4-5), CD8α (53–6.7). For intracellular FoxP3 or Ki67 staining, cell samples were fixed, permeabilized, and stained with anti-mouse Foxp3 (FJK-16s). All fluorescent-labeling mAbs were purchased from BioLegend or eBioscience. H-2K$^b$ SIY and H-2K$^b$ OT-I tetramers were purchased from MBL (Japan). Antibodies used in this study had been listed in Supplementary Table 1. DAPI or LIVE/DEAD™ fixable yellow dye (ThermoFisher) was used to exclude dead cells. Samples were analyzed on a FACSCalibur (BD) or FACS Fortessa flow cytometer (BD). Data were analyzed using FlowJo software (TreeStar).

**Ex vivo-binding assay**. A single-cell suspension of Foxp3-GPF transgenic mice (BALB/c background) spleen cell (1*10e6) was incubated with anti-CD16/32 (anti-FcγIII/II receptor, clone 2.4G2) for 30 min at 4 °C. WT IL-2-Fc, F42A IL-2-Fc, super IL-2-Fc, sumIL2-Fc, and hIgG were added to the cells for 20 min on ice, washed twice, and then stained with anti-human IgG Fcγ-PE for 20 min on ice. After washing twice, samples were analyzed on the FACS Fortessa flow cytometer (BD). Data were analyzed using FlowJo software (TreeStar).

**Fluorescence imaging**. Erb-sumIL2 was labeled with Cy5.5, purified. The binding ability was detected by FACS. Fluorescently-labeled Erb-sumIL2 (25 μg of antibody) was injected intravenously into C57BL/6 mice bearing subcutaneous MC38 (left flank) and MC38-EGFR5 (right flank) tumors. Fluorescence was measured with IVIS Spectrum at different time points.

**Toxicity**. Mice were killed on the 7th day after injections of 25 μg of Erb-sumIL2 for three times. Pulmonary wet weight was measured to assess adverse toxic effects following sumIL-2 treatment and was determined by weighing the lungs before and after drying.

**Statistical analysis**. All analyses were performed using GraphPad Prism statistical software (GraphPad Software Inc., San Diego, CA). Mean values were compared using an unpaired Student's two-tailed t-test. $P < 0.05$ was considered statistically significant.

**Reporting summary**. Further information on research design is available in the Nature Research Reporting Summary linked to this article.

## Data availability
The data that support the findings of this study are available from the corresponding author on reasonable request.

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

## Acknowledgements

We are grateful to Dr Mingzhao Zhu (Institute of Biophysics, CAS) for helpful suggestions and comments on the project. We thank Dr Yihui Xu and Dr Xiaoyan Wang for their technical assistance. We appreciate the funding from the Chinese Academy of Sciences (KFJ-STS-ZDTP-062) and (XDA12020212) to H.P., National Key S&T Special project of China (2018ZX1030140402) to H. Peng.

## Author contributions

Conception and design: Z.S., H.P., and Y.-X.F., Development of methodology: Z.S., Acquisition of data: Z.S., Z.R, K.Y., S.D., S.C., Y.L., and J.G., Analysis and interpretation of data (e.g., statistical analysis, biostatistics, computational analysis): Z.S., Y.-X.F., and H.P., Writing, review, and/or revision of the manuscript: Z.S., L.X., H.P., and Y.-X.F., Administrative, technical, or material support (i.e., reporting or organizing data, constructing databases): H.X., J.S., F.W., H.P., and Y.-X.F., Study supervision: H.P., Y.-X.F.
