## [Peer Review File · Nature Communications]

Reviewers' Comments:

Reviewer #1:

Remarks to the Author:

In this manuscript the authors have addresses a critical issue in tumor immunology, namely the challenges of supplying IL-2 to tumor sites to promote TIL growth and function. As highlighted by the authors, typically tumor infiltrating CD8 cells are compromised by lack of IL-2. As an approved therapeutic, treatment of patients with IL-2 is associated with better clinical responses in some patients, however, the toxicity and yin-yang effects of promoting Tregs in the tumor site have dramatically limited its utility. In the present study, the authors confirm in animal models that IL-2 is limiting at the tumor site, low dose and even IL-2Fc does not overcome this limitation. The quantitation of IL-2 at the site is not convincing and the dosing was not studied in detail but the conclusions that wild type IL-2 and lower CD25 affinity F42A did not enhance tumor immunity sufficiently to cure mice. In addition the creation of a super mutant that combined F42A and other mutations previously published to increase CD122 binding moderated relevant affinities. The data are convincing (especially in light of previous published results that the novel sumIL-2-Fc). However, only our sumIL-2 favorably binds selectively to activated CD8 T cells but less so to Treg cells. These molecule also led to increased anti-tumor activity consistent with previous studies. The major concern in this protein is the likelihood that with all the mutations and activity in boosting immunity, that the drug will likely be immunogenic and thus have limited usefulness in human. This would be difficult to test in mouse models and humanized mouse models are lacking for good assessment of immunogenicity. This is a problem in the field in general.

The investigators hypothesize that in order to avoid systemic side effects, the sumIL-2-Fc would be safer and more efficacious if it could be localized to the tumor site. This led them to propose the development of a bispecific reagent that simultaneously recognized human EGFR using a monomer of an anti-EGFR mAb and a bound molecule of sumIL-2. The investigators show that the intratumoral injection of ERb-sumIL-2 is more efficacious than systemic administration supporting the hypothesis. Functional testing of the ERb-sumIL-2 suggested that the drug localized better to tumor sites expressing mutated mouse EGFR and treatment led to better efficacy. There are two concerns with the current studies. There were no controls addressing whether the tumors expressing the mutated mouse EGFR was more immunogenic influencing the effects of IL-2. The fact that the F42A did not have a similar effect may be because Tregs were induced. In addition, the investigators did not test the drug on EGFR negative tumors. Both of these points need to be addressed.

The rest of the paper is a useful but straightforward set of studies suggesting the major effect of the localized therapy was on local CD8 T cells in the tumor. In addition, the studies suggest that in the mouse model, the drug synergizes with other therapies like TKIs and checkpoint inhibition giving some hope that the combination may be more effective in patients. Finally, the study suggests that when given at the time of primary tumor removal, the drug can be efficacious in reducing metastases. These are all useful observations but in pretty limited models which may or may not be translatable.

Overall, this is a worthwhile study and should be published but the controls requested above should be performed.

Reviewer #2:

Remarks to the Author:

In this study, the authors constructed F42A and H9 based IL-2 (sumIL-2) fusion proteins and found that the sumIL-2 has not only increased half-life, but also enhanced treatment effect by better targeting CD8+ T cells inside tumor tissues. SumIL-2 can also overcome the resistance to TKI treatment. Furthermore, sumIL-2 can function as an "immune accelerator" to enhance the

therapeutic effects of checkpoint blockades.

Comments on improving the manuscript:

1. SumIL-2 are IL-2 mutations from the combination of F42A and H9 (including the mutation of L80F, R81D, L85V, I86V and I92F), for the rationale of taking advantage of increasing IL-2 binding to T cells of Super IL-2 (H9) and decreased IL-2 binding to Treg of F42A and the data confirms it (Figure 1D-1F). However, there was no proof that this difference resulted in treatment benefit in any animal tumor models they tested, since there was no SuperIL2-Fc control in any of these experiments. At least in Fig 1I, this control should be included.

2. In this study, the data shows that i.v. delivery of Erb-sumIL2 improves treatments by targeting EGFR-expressing tumors. While most of the work in this study use local injections of either Erb-sumIL2 or sumIL2-Fc, there is no experiment comparing their equivalency or difference. Therefore, it is confusion for why sometime use Erb-sumIL2 and sometime use sumIL2-Fc, such as in Figure 7, A used ab-sumIL2, while B used SumIL2-Fc.

3. Figure 1C only shows data from reference publication.

4. Statistical analysis is required for Fig 1A.

5. What's SIY stands for? What's FTY720 stands for?

6. In EGFR negative B16-SIY and MC38-OVA models, Erb-sumIL2 treatment showed increased SIY or OVA reactive T cells, the mechanisms of how Erb-sumIL2 induce tumor-specific T cell response should be explained.

Reviewers' comments:

Reviewer #1 (Remarks to the Author):

Overall, this is a worthwhile study and should be published but the controls requested above should be performed.

The major concern in this protein is the likelihood that with all the mutations and activity in boosting immunity, that the drug will likely be immunogenic and thus have limited usefulness in human. This would be difficult to test in mouse models and humanized mouse models are lacking for good assessment of immunogenicity. This is a problem in the field in general.

We agree with the reviewer that this is a difficult problem in the field in general. Whether mutated IL-2 can be immunogenic protein in human is difficult one to test. We have explored the immunogenicity of human IL-2. We propose that mouse and human IL-2 have very similar in structures which make human IL-2 poor immunogenic in mouse especially through short course of intratumoral or i.v. delivery. SumIL-2 contains very few mutant amino acids which may not have major increase of immunogenicity of human IL-2 in mouse system. To assess the immunogenicity of sumIL-2, we expressed and purified sumIL-2 with His tag and found no detectable anti-sumIL-2 antibody after sumIL2-Fc treatment (Figure R1, left). However, anti-human IgG Fc can be detected in the serum (Figure R1, right). These data suggested that a short course and systemic delivery of sumIL-2 in mice could not generate detectable antibody responses for human IL-2.

Figure R1. Undetectable antibody responses to human IL-2 in C57BL/6 mice after human IL-2 delivery. C57BL/6 mice (n=7-8/group) were injected subcutaneously with 5×10^5 B16F10 tumor cells, and $5 \mu\text{g}$ of sumIL2-Fc was administered intratumorally (i.t.) on day 9. Serum was collected on day 23. Anti-sumIL2-His and anti-human IgG Fc in serum were tested by ELISA (serum dilution ratio was 1:100).

There are two concerns with the current studies. There were no controls addressing whether the tumors expressing the mutated mouse EGFR was more immunogenic influencing the effects of IL-2. The fact that the F42A did not have a similar effect may be because Tregs were induced. In addition, the investigators did not test the drug on EGFR negative tumors. Both of these points need to be addressed.

Thanks for the reviewer's comments. EGFR5 has all mouse sequences except six amino acids replacing for Cetuximab binding. There has no detectable increase of immunogenicity.

In both EGFR5 positive and negative tumor lines, we consistently observed more potent effect of sumIL2-Fc than WT IL-2-Fc. And for the mechanism study, we also used EGFR5 negative tumor to study the cellular immune responses induced by IL-2 variants. In addition, we set up a double-tumor model. In figure 4E, WT C57BL/6 mice were injected subcutaneously with MC38 (left flank) and MC38-EGFR5 cells (right flank). MC38-EGFR5 tumors were i.t. treated with Cetuximab (Erb) or Erb-sumIL2. The growth of MC38 tumor on another site was also inhibited, this data suggested that the anti-tumor response induced by sumIL-2 was MC38-specific, but not depending on the potential immunogenicity of EGFR. Moreover, we detected whether TILs were different between mutant EGFR positive and negative tumor in the double-tumor model. We actually found that TILs were higher in MC38 than in MC38-EGFR5 tumor, it suggested mutant EGFR did not induce more TILs (Figure R2). The data support the therapeutic effects of IL-2 were not influenced by mutant EGFR.

Figure R2. WT C57BL/6 mice (n=6/group) were injected subcutaneously with 5×10^5 MC38 (left flank) and MC38-EGFR5 (right flank) cells. Tumor tissues were collected on day 9 and The percentages of CD45⁺ among live cells (left) , CD4⁺ among CD45⁺ cells (middle) and CD8⁺ among CD45⁺ cells (right) were analyzed by flow cytometry.

We also detect whether Treg cells were still induced after F42A IL-2 treatment. Indeed, comparing to sumIL-2, F42A IL-2 couldn't increase the frequency of CD3⁺ T cells, but it still

increases the frequency of Treg cells. These might limit the therapeutic effects of F42A IL-2 (Figure R3).

Figure R3. WT C57BL/6 mice were injected subcutaneously with 5×10^5 MC38-EGFR5 cells and i.t. treated with $10 \mu\text{g}$ of Erb-F42A IL2 or Erb-sumIL2 on day 9. The tumor tissue were collected on day 12, and the percentages of CD3+ among total live cells (A) and Treg among CD4+ cells (B) were analyzed by flow cytometry.

Reviewer #2 (Remarks to the Author):

In this study, the authors constructed F42A and H9 based IL-2 (sumIL-2) fusion proteins and found that the sumIL-2 has not only increased half-life, but also enhanced treatment effect by better targeting CD8+ T cells inside tumor tissues. SumIL-2 can also overcome the resistance to TKI treatment. Furthermore, sumIL-2 can function as an “immune accelerator” to enhance the therapeutic effects of checkpoint blockades.

Comments on improving the manuscript:

1. SumIL-2 are IL-2 mutations from the combination of F42A and H9 (including the mutation of L80F, R81D, L85V, I86V and I92F), for the rationale of taking advantage of increasing IL-2 binding to T cells of Super IL-2 (H9) and decreased IL-2 binding to Treg of F42A and the data confirms it(Figure 1D-1F). However, there was no proof that this difference resulted in treatment benefit in any animal tumor models they tested, since there was no SuperIL2-Fc control in any of these experiments. At least in Fig1I, this control should be included.

We thank the reviewer’s comments and have compared the anti-tumor effects of tumor-targeting IL-2 variants. In figure below, SumIL-2 can control tumor more effectively than WT IL-2 and other mutant IL-2 (Figure R4). And we have used this figure to replace the previous Figure S4E (line 186-188)

Figure R4. C57BL/6 mice (n=5/group) were injected subcutaneously with 5×10^5 MC38-EGFR5 cells, and then i.p. treated with 25 μ g of Erb-IL2 variants on day 9 and the tumor volume was measured.

2. In this study, the data shows that i.v. delivery of Erb-sumIL2 improves treatments by targeting EGFR-expressing tumors. While most of the work in this study use local injections of either Erb-sumIL2 or sumIL2-Fc, there is no experiment comparing their equivalency or difference. Therefore, it is confusion for why sometime use Erb-sumIL2 and sometime use sumIL2-Fc, such as in Figure 7, A used ab-sumIL2, while B used SumIL2-Fc.

We thank the reviewer for pointing out the potential issue to be clarified. We will explain local and systemic delivery better in our revision. We intratumorally injected Erb-sumIL2 and sumIL2-Fc respectively, containing comparable molar amount of sumIL-2. Similar anti-tumor effects were observed with either i.t. 10 μ g Erb-sumIL2 (1 x sumIL2) treatment or 5 μ g sumIL2-Fc (2 x sumIL2) treatment, suggesting the comparable potency of both Erb-sumIL2 and sumIL2-Fc when locally delivery (Figure R5). However, we have shown that local delivery of Ab-sumIL2 is far more potent than systemic delivery (Figure 2B). Then, we demonstrated tumor-targeting effect by systemic delivery of Erb-sumIL2 to EGFR5 (+) tumor (Figure 2C). We used systemic delivery of Ab-IL2 (i.v. or i.p. injection) to compare

the tumor targeting and therapeutic efficacy, while we use intratumoral treatment mainly to study whether and how IL2 might directly regulate TILs relatively independent on the effect of IL-2 outside tumor microenvironment.

Figure R5. WT C57BL/6 mice (n=5/group) were injected subcutaneously with 5×10^5 of B16F10 cells and i.t. treated with 10 μ g of Erb-sumIL2 or 5 μ g of SumIL2-Fc on days 9, 12 and 15. Tumor volume was measured at the indicated time points.

For Figure 7A, we found that i.v. Erb-sumIL2 treatment resulted in upregulation of PD-L1 in DCs which may inhibit DC function and T cell activation. We then tested the anti-tumor effect of Erb-sumIL2 in combination with the checkpoint blockade anti-PD-L1 (Figure 7B). Our data demonstrate that Erb-sumIL2 and anti-PD-L1 have synergistic anti-tumor function. To make a clear presentation and consistency with Figure 7A, we have replaced the combination therapy result of i.t. sumIL-2 treatment with data of i.p. Erb-sumIL2 treatment in Fig 7B (Figure R6).

Figure R6. WT C57BL/6 mice (n=5/group) were injected subcutaneously with 5×10^5 of B16F10-EGFR5 cells and i.p. treated with 25 μ g of Erb-sumIL2 or/and i.t. treated 50 μ g of anti-PD-L1 on days 8, 11 and 14. Tumor volume was measured on the indicated time points.

3. Figure 1C only shows data from reference publication.

Thank the reviewer for pointing out this concern. This data was not from the reference publication. We had further analyzed the raw data online and made a new conclusion which was not analyzed or mentioned in the original reference. We have revised the manuscript as “We also analyzed the expression level of CD25 on different subtypes of T cells in the cancer patients based on the single cell RNA-sequencing data from Zhang’s group. The result showed that the expression level of CD25 on Treg cells was much higher than effector T cells in the cancer patients (Fig 1C) (Line114-117)”.

4. Statistical analysis is required for Fig 1A.

Statistical analysis has been added in Fig 1A.

5. What’s SIY stands for? What’s FTY720 stands for?

SIY (SIYRYGL) is a synthetic peptide associated with the class I molecule K^b which can be recognized by 2C TCR transgenic T cell. In B16-SIY model, SIY as an antigen can be presented by SIY expressing tumor cell or cross-presented by APCs expressing K^b. We use associated tetramer to detect SIY-specific CD8⁺ T cells. We have added this introduction in

the manuscript (Line222-227).

FTY720 is a structural analogue of the sphingosine-1-phosphate (S1P) and is derived from myriocin. It can potently inhibit the egression of T cells from the LN into the circulation and peripheral tissues. We have added this into new introduction in the manuscript (Line251-254).

6. In EGFR negative B16-SIY and MC38-OVA models, Erb-sumIL2 treatment showed increased SIY or OVA reactive T cells, the mechanisms of how Erb-sumIL2 induce tumor-specific T cell response should be explained.

We thank the reviewer for pointing out this confusion.

It has been shown that a large amount of tumor-specific T cells inside tumor tissues are dysfunctional. We propose that lack of IL-2 limits proper expansion of antigen-specific T cells (SIY-reactive T cells in B16-SIY model and OVA-reactive T cells in MC38-OVA model), and preexisting antigen-reactive T cells can be re-activated and expanded when IL-2 is sufficiently provided. Here, we used i.t. injection of Erb-sumIL2 into EGFR negative B16-SIY and MC38-OVA tumor to assess the change of antigen-specific T cell inside tumor, but not for the targeting delivery of sumIL-2. We have now added the explanation into the revision (Line234-240).

Reviewers' Comments:

Reviewer #1:

Remarks to the Author:

accept

Reviewer #2:

Remarks to the Author:

All questions and concerns are properly addressed. There is no further questions.